# Ag_2_S QDs/Si Heterostructure-Based Ultrasensitive SWIR Range Detector

**DOI:** 10.3390/nano10050861

**Published:** 2020-04-29

**Authors:** Ivan Tretyakov, Sergey Svyatodukh, Aleksey Perepelitsa, Sergey Ryabchun, Natalya Kaurova, Alexander Shurakov, Mikhail Smirnov, Oleg Ovchinnikov, Gregory Goltsman

**Affiliations:** 1Astro Space Center, Lebedev Physical Institute of the Russian Academy of Sciences, Moscow 117997, Russia; 2Institute of Physics, Technology, and Informational Systems, Moscow Pedagogical State University, Moscow 119435, Russia; sergey.svetodux@gmail.com (S.S.); a-perepelitsa@yandex.ru (A.P.); sergey.ryabchun@gmail.com (S.R.); nkaurova@yandex.ru (N.K.); alexander@rplab.ru (A.S.); goltsman@mspu-phys.ru (G.G.); 3Faculty of Physics, Voronezh State University, Voronezh 394018, Russia; smirnov_m_s@mail.ru (M.S.); ovchinnikov_o_v@rambler.ru (O.O.); 4School of foreign languages, National Research University Higher School of Economics, Moscow 101000, Russia; 5Scientific and Educational Center “NanoBioTech”, Voronezh State University of Engineering Technologies, Voronezh 394017, Russia; 6LLC “Superconducting Nanotechnology” (Scontel), Moscow 119021, Russia; 7Laboratory of nonlinear optics, Zavoisky Physical-Technical Institute of the Russian Academy of Sciences, Kazan 420029, Russia

**Keywords:** short-wave infrared range, silicon, quantum dots, detector

## Abstract

In the 20^th^ century, microelectronics was revolutionized by silicon—its semiconducting properties finally made it possible to reduce the size of electronic components to a few nanometers. The ability to control the semiconducting properties of Si on the nanometer scale promises a breakthrough in the development of Si-based technologies. In this paper, we present the results of our experimental studies of the photovoltaic effect in Ag_2_S QD/Si heterostructures in the short-wave infrared range. At room temperature, the Ag_2_S/Si heterostructures offer a noise-equivalent power of 1.1 × 10^−10^ W/√Hz. The spectral analysis of the photoresponse of the Ag_2_S/Si heterostructures has made it possible to identify two main mechanisms behind it: the absorption of IR radiation by defects in the crystalline structure of the Ag_2_S QDs or by quantum QD-induced surface states in Si. This study has demonstrated an effective and low-cost way to create a sensitive room temperature SWIR photodetector which would be compatible with the Si complementary metal oxide semiconductor technology.

## 1. Introduction

Over the past few decades, silicon (Si) has been the main technological semiconducting material and has determined the evolution of our society [1]. The development of efficient room temperature photoconductors for the infrared (IR) range that are compatible with modern silicon technology is in great demand. In particular, megapixel digital imaging based on the complementary metal oxide semiconductor [2] technology in the IR range has a high potential for important technological applications, such as quantum-cryptographic key distribution, quantum communication, on-chip data processing, night vision, IR spectroscopy, medical diagnostics, environmental monitoring, and astronomy [3,4,5,6,7,8,9,10,11,12,13,14,15]. Thus, the problem of expanding the absorption range of Si to the short-wave infrared range (SWIR) is of considerable fundamental and practical importance.

It is known that by creating impurity states in the band-gap of Si, one can control the process of sub-band absorption of radiation. Modern techniques involve expanding the spectral range of the response of Si with the use of either high-temperature doping with chalcogens such as S, Se, Te, or metals such as Au, Ag, and Ti [16,17,18,19]. These methods are not applicable to the existing technology of manufacturing silicon microcircuits based on the complementary metal oxide semiconductor (CMOS) because of the need to heat the silicon to extremely high temperatures during ion implantation. Another approach is based on the integration of non-silicon electro-optical materials, such GaAs, InAs. This approach is strongly limited by the mismatch between the lattice of Si and the lattice of these materials; however, significant progress has recently been made in the fabrication and study of high-quality structures based on materials from groups III–V grown on an Si substrate [20,21,22,23]. Commercially available Si:As impurity band conduction detectors are widely used for the detection of IR radiation with wavelengths of 5–28 um [24,25]. Two- and one-dimensional materials do not suffer from this limitation and can be transferred to any substrate and used for radiation detection in the IR region [26,27,28,29]. A chemical vapour deposition grown (CVD-grown) graphene monolayer coated with colloidal quantum dots (QDs) [30] has been monolithically integrated with Si CMOS for digital imaging in the SWIR [4]. Besides, silicon integrated electro-optical materials, high-quality detectors based on HgCdTe and InSb are commercially available to detect IR at wavelengths of 0.8–2.5 um and 0.8–5.4 um, respectively [31,32].

In our previous paper Ref [33], we experimentally demonstrated the possibility of expansion of the absorption of high-resistance uncompensated silicon in the SWIR region through the deposition of colloidal Ag_2_S quantum dots with a size of 2.5 nm on its surface. Based on the experimentally obtained spectral dependence of the response of the Ag_2_S/Si structure, a hypothesis was proposed about possible mechanisms of the photovoltaic effect at wavelengths greater than 1.1 μm. Understanding physical mechanisms behind the photovoltaic effect greatly facilitates the development of devices based on QDs/Si structures with ultimate detection capabilities. It is known that the optical properties of QDs strongly depend not only on their material, but also on their size [34,35,36,37]. Thus, the sensitivity of QD-based devices at some point will be limited by their properties. In this work, to understand the mechanisms of the photovoltaic effect in the Ag_2_S/Si structure, we study experimentally the spectral dependences of the response of structures of coated Ag_2_S with sizes of 1.8 nm, 2 nm, and 3.5 nm.

## 2. Ag_2_S QDs Preparation and Experimental Methods

In our experiments we used Ag_2_S QDs prepared in a water solution of thioglycolic acid (TGA). The initial reagents used in the work were Na_2_S, AgNO_3_, TGA, and high-purity NaOH (Sigma Aldrich, St. Louis, MO, USA). The size variation was carried out by changing the ratio Ag^+^:S_2_^−^. In particular, for the preparation of QDs with an average size of 1.8–2.0 nm, the passivator molecules TGA acted as a sulfur precursor, and the Ag^+^:S_2_^−^ ratio changed from 1:0.9 to 1:1. To obtain larger QDs, an additional sulfur precursor (Na_2_S) was used. For this, a solution of the Ag^+^/TGA precursor (200 mL) was used, which was obtained by mixing an AgNO_3_ solution (2.6 mmol) and TGA (2.6 mmol), followed by adjusting the pH to 10 with a 1 M NaOH solution. Then, 50 mL of a water solution of Na_2_S was added to the prepared solution with constant stirring using a peristaltic pump. The solution in the reactor changed colour from pale yellow to dark brown. Thus, the formation of Ag_2_S/TGA QDs with a concentration in water of 2 × 10^−5^ mol CT/L was realized. The synthesis was carried out at a temperature of 30 °C. The ratio of Ag^+^:S_2_^−^ in this case was 1:0.33. The synthesis technique is described in more detail in [38].

The structural studies of Ag_2_S QDs were carried out by the method of transmission electron microscopy (TEM), including high-resolution TEM (HR-TEM). The sizes of the synthesized ensembles of colloidal QDs were determined using a Libra 120 PLUS transmission electron microscope with an accelerating voltage of 120 kV (Carl Zeiss, Oberkochen, Germany). The size distributions for the QD ensemble were obtained by the digital analysis of TEM images. HR-TEM images of Ag_2_S QDs were obtained using a JEM-2100 transmission electron microscope (Jeol, Akishima, Tokyo, Japan) with an accelerating voltage of 200 kV. X-ray diffraction (XRD) patterns were obtained with an ARL X’TRA diffractometer (Thermo Fisher Scientific, Waltham, MA, USA) for Cu Kα1 (1.54 Å) radiation.

The obtained Ag_2_S QDs, according to the TEM data (see the left part of Figure 1), had average sizes of 1.8, 2.0, and 3.2 nm with a size dispersion of 7–25%. The high-resolution TEM images show that the presence of interatomic planes corresponds to the monoclinic crystalline lattice of silver sulfide (P2_1_/c). The XRD patterns of small Ag_2_S QDs (1.8 and 2.0 nm) presented in Figure 1 show a wide halo at 20–50°. This is characteristic of the low-ordered monoclinic crystal lattice of Ag_2_S, the presence of which was established with the use of HR-TEM. The diffuse nature of the XRD pattern is due to the abundance of crystallographic planes in the monoclinic lattice and the corresponding reflections on the X-ray diffraction pattern, the broadening of which as a result of the size effect leads to the formation of a wide structureless band [39,40,41,42]. For Ag_2_S QDs with average size of 3.2 nm, the formation of low-intensity peaks in diffraction patterns was found, the position of which corresponds to reflections of the Ag_2_S monoclinic crystal lattice.

Optical absorption was obtained with the use of USB2000+ spectrometer with an USB-DT radiation source (Ocean Optics, Dunedin, FL, USA). In the optical absorption spectra of our samples presented in Figure 2, we observe broad bands in a range of 1.5–5 eV with characteristic features at of 1.6 eV, 2.1 eV, and 2.7 eV, for Ag_2_S QDs with an average size of 3.2 nm, 2.0 nm, and 1.8 nm, respectively. The appearance of these features is associated with the predominance of ground state exciton absorption in the spectra. The presence of a smooth absorption edge in the region of lower energies for the studied samples could be caused by two factors—(i) size dispersion and (ii) the impurity absorption of light by trap states, the presence of which is caused by the non-stoichiometry of Ag_2_S [43].

Photoluminescence spectra in the spectral range of 500–1000 nm were obtained with the use of USB2000+ spectrometer (Ocean Optics, Dunedin, FL, USA). Photoluminescence spectra were obtained with an automated spectrographic instrument based on an MDR-4 diffraction monochromator (LOMO, Saint Petersburg, Russia). A highly stable low-noise PDF10C/M photodiode (Thorlabs Inc., Newton, NJ, USA) with a built-in amplifier was used as the photodetector in the near-IR region. An NDB7412T-1W laser diode (Nichia, Anan, Tokushima, Japan) emitting at a wavelength of 445 nm with an optical power of 440 mW was used as the excitation source.

The Ag_2_S QDs used have size-dependent photoluminescence in the region of 600–1100 нм (Figure 3). The maxima of luminescence bands are at 660 nm, 620 nm, and 950 nm for Ag_2_S QDs with average size of 1.8 nm, 2.0 nm, and 3.2 nm, respectively. The nature of the luminescence is different. QDs with size 1.8 nm and 3.2 nm have trap-state luminescence. Ag_2_S QDs with an average size of 2.0 nm show characteristic exciton luminescence. Conclusions about the nature of the luminescence bands were made based on the analysis of the band parameters (Stokes shift, half-width (FWHM)), luminescence decay time, etc. [44].

A few groups of shallow trap states with energy depths of 0.06–0.12 eV involved in non-radiative carrier capture were detected in Ag_2_S QDs with different sizes by the method of thermally stimulated luminescence (TSL), which is detailed and described in [45]. In the TSL modification used by us, the sample with constant photoexcitation was cooled to the temperature of liquid nitrogen, where it was kept for some time (about 5 min); then, it was heated to the initial temperature. Throughout the experiment, photoluminescence intensity was recorded. Next, the difference between the temperature dependences obtained by heating the sample and its cooling was calculated. The resulting difference is due to the thermal release of charge carriers localized to trap states. Then, using the kinetic model of the process, the depths of the corresponding trap states were determined [45].

The substrates are high-resistance silicon wafers with ρ > 3 kΩ·cm and a thickness of 350 μm. The Ag_2_S/Si heterostructure is formed in the space between two Ti/Au contacts on the Si substrate coated with Ag_2_S QDs. In our case, the width and the length of the gap are 10 μm. Figure 4a shows an optical image of the inner part of the heterostructure.

The Ti/Au contacts to Si are made by laser lithography and the lift-off process, followed by the thermal deposition of 5 nm of Ti and 200 nm of Au through the resist windows onto a previously cleaned Si surface. The cleaning of the Si surface is done in two stages: ion etching in oxygen for 15 s at 50 W and in argon for 20 s at 50 W. Liquid etching with hydrofluoric acid (HF:H_2_O = 1:10) takes 30 s. Figure 4c shows a typical IV curve of one our Au/Ti/Si/Ti/Au structures. The shape of the IV curve indicates the formation of an asymmetric spatial charge distribution at the Si/Ti interface. Subsequently, the presence of the Schottky barrier makes it possible to eliminate the need for mixing devices with an external electric field when studying their spectral characteristics.

The fabrication of Ag_2_S/Si heterostructures must not lead to the formation of a potential barrier at the interface of Ag_2_S and QDs; therefore, before the deposition of QDs, the surface of Si in the gap between the metal contacts is also cleaned, as in the fabrication of Ti/Au contacts to Si. The Ag_2_S QDs are deposited by spin-coating for 1 min at 2000 rpm on top of the cleaned Au/Ti/Si/Ti/Au structures heated to 120 °C. The temperature of 120 °C used in the deposition of Ag_2_S QDs on the silicon surface was selected in order to preserve the phase structure of the Ag_2_S monoclinic crystal lattice. The Ag_2_S phase transition from a monoclinic modification to a cubic body-centered (bcc) occurs at a temperature of 176 °C [46], which is significantly higher than the values used by us. That is confirmed by XRD analysis (Figure 1, Curves 1′–3’). The deposition of Ag_2_S QDs on an Si substrate leads to the appearance of reflections with maxima in the regions of 29.3°, 32.9°, and 69.1° corresponding to the crystallographic planes of a silicon single crystal.

We assess the quality of the fabricated samples by comparing the spectral dependences of the photovoltaic response obtained for Ag_2_S/Si heterostructures and a sample of uncoated structures in the wavelength range 1–2 μm. Spectral measurements are made with a monochromator with a line width of 5 nm.

## 3. Results and Discussion

The IR radiation from the monochromator is focused with an elliptical silicon lens into an airy spot on the Si surface between the Ti/Au contacts of the device. Referring to Figure 4b, the heterostructure mounted on the lens is placed in a vacuum holder equipped with optical filters and electrical connectors; the filters provide a pass band of 1–2.5 μm. The photovoltaic response of the samples to the amplitude-modulated, collimated radiation is read-off with a lock-in amplifier.

Figure 5 shows a family of spectral dependences for Ag_2_S/Si heterostructures with different diameters of Ag_2_S QDs, as well as the spectral dependence for a few uncoated structures. The graph shows that at wavelengths greater than 1.2 µm, the responsivity of Ag_2_S/Si heterostructures where QDs are 1.8 nm and 3.2 nm in size significantly exceeds that of the uncoated heterostructures, and that responsivity grows with the size of the QDs. According to the experimental data of absorption optical spectroscopy, the energy of ground state exciton absorption is 2.70 +/− 0.01 eV and 1.6 +/− 0.01 eV for Ag_2_S QDs with an average size of 1.8 and 3.2 nm, respectively. Therefore, the response of the Ag_2_S/Si heterostructures cannot be caused by direct exciton absorption. The non-stoichiometry of Ag_2_S QDs is known to cause a high concentration of defects in the crystalline structure of QDs and, as a consequence, a rich spectrum of trap states. Trap states lead to the appearance of an extended tail in the SWIR region of the absorption spectrum of QDs. Since the band gap of Ag_2_S QDs decreases with increasing the diameter of the dots, the farther the tail extends into the SWIR region, the larger the diameter of the QDs. Thus, the mechanism of the photoresponse beyond the intrinsic absorption band of Si and excitonic Ag_2_S QDs consists in the appearance of diagonal transitions between the valence band of silicon and electronic levels of dimensional quantization of Ag_2_S QDs involving trap states. The TSL data show a change in the structure of the trap states as a result of the deposition of Ag_2_S QDs on the surface of the Si substrate. In particular, the formation of new thermal emission peaks, which are not characteristic of colloidal solutions of Ag_2_S QDs ensembles, is observed. So, for Ag_2_S QDs with an average size of 3.2 nm, a transformation of the thermal emission curve is observed after the deposition on a Si substrate (Figure 6). As a result of the deposition, a decrease in the thermal emission intensity of Ag_2_S QDs deposited on Si is observed in comparison with the initial QDs solution, which indicates the formation of new non-radiative carrier transfer channels between the components of the heterostructure. Moreover, the number of Ag_2_S QDs deposited on Si substrates was equivalent to the number of Ag_2_S QDs in the colloidal solution of the reference sample. According to our estimates, the occurrence of these thermal emission peaks are responsible for trap states with energy depths of 0.055 eV and 0.12 eV in Ag_2_S/Si heterostructure and 0.11 eV in the colloidal solution 3.2 nm Ag_2_S QDs. The detected changes in the energy structure of the trap states are most likely due to the formation of new charge transfer channels between Ag_2_S QDs and Si substrate as a result of the formation of the heterostructures under discussion. These channels are capable of providing light absorption outside the intrinsic absorption bands of Si and Ag_2_S QDs, which we observed in our measurements of spectral response curves of prepared heterostructures.

The spectral dependence of the response of the Ag_2_S/Si heterostructures with a QD size of 3.2 nm has a noticeable kink in the region of 1.45 μm, which may indicate a change in the mechanism of the photovoltaic effect. To study the nature of the kink, we have performed a detailed study of the response of Ag_2_S/Si heterostructures with a QD size of 1.8 nm and 2 nm at wavelengths of 1.31 μm and 1.55 μm. The choice of the size of QDs is determined by the need of complete exclusion or significant suppression of detection because of the SWIR “tail” QDs at λ 1.55 μm. The use of stabilized fiber-optic laser diodes operating at 1.31 μm and 1.55 μm as radiation sources makes it possible to see the difference in the response of Ag_2_S/Si heterostructures with a QD size of 1.8 nm and 2 nm, which is hardly distinguishable in broadband spectral measurements. During the measurements, a linear dependence of the response value on the radiation power of the laser diode is checked and the power is chosen such that the signal-to-noise ratio does not exceed 2–3. Figure 7 shows the experimental values of the voltage responsivity of the Ag_2_S/Si heterostructures at wavelengths of 1.31 μm and 1.55 μm obtained for the Ag_2_S/Si heterostructures. The upper panel of Figure 7 shows that the voltage responsivity measured at λ 1.31 μm for heterostructures with a QD size of 2 nm is noticeably greater than for heterostructures with a QD size of 1.8 nm. At the same time, within the experimental error, the voltage responsivity measured at 1.55 μm for the same samples (the bottom panel of Figure 7) does not depend on the QD size. Therefore, the response mechanism of the Ag_2_S/Si heterostructures at 1.31 μm is determined by the size of QDs, and at 1.55 μm, it is practically independent of the size of the Ag_2_S QDs. In addition, the voltage responsivity of the Ag_2_S/Si heterostructures at 1.55 μm is much greater than that of the Si structure not coated with QDs (Figure 5). As is seen from our measurements, at 1.55 μm, QDs do not directly absorb radiation, but, being located on the Si surface, they significantly enrich the spectrum of the surface states. Taking into account the mobility of the carriers in high-resistance silicon [47], the surface states formed by Ag_2_S QDs can create additional vacancies for electrons from the valence band of silicon. This makes it possible to detect photons with a wavelength greater than 1.55 μm.

For the Ag_2_S/Si heterostructures with high voltage responsivity, we have measured the noise level and calculated the noise-equivalent power (NEP). The presence of built-in potential barriers near the Ti/Si contacts allows us to carry out spectral measurements at zero bias of the Ag_2_S/Si heterostructures. However, the uncontrolled height ratio of these Schottky barriers may significantly lower the voltage responsivity of Ag_2_S/Si heterostructures. For this reason, the voltage responsivity of the detector based on Ag_2_S/Si heterostructures is measured when the heterostructures are biased in the voltage mode. Figure 8 shows the dependences of the current flowing through Ag_2_S/Si, the resistance of Ag_2_S/Si, and the recorded signal as a function of the bias voltage.

The voltage dependence of the signal shows that at a bias of 0.4 V, the signal increases more than 20-fold compared to the signal at zero bias. The dependence correlates well with the dependence of the sample resistance on the voltage, while the maximum responsivity is achieved in the region with the maximum slope of the IV curve of the heterostructure, which is typical of diodes with a Schottky barrier in the millimeter (MM) and submillimeter (subMM) ranges. The best NEP for the Ag_2_S/Si heterostructures with a QD size of 3.2 nm and a surface density of 10 QDs per 10 nm^2^ is 1.1 × 10^−10^ W/√Hz at room temperature, which is better compared to the best commercially available room temperature SWIR detectors.

## 4. Conclusions

In conclusion, this study has demonstrated an effective and low-cost way to create a sensitive room temperature SWIR photodetector compatible with the Si CMOS technology. Based on the results of spectral response measurements of Ag_2_S/Si heterostructures, two main mechanisms of photoresponse in the SWIR region can be distinguished: the effective absorption of SWIR photons can be either due to the QD crystal lattice defects or due to quantum QD-induced surface states in Si. The Ag_2_S/Si heterostructure with a sensitive area of 10 × 10 μm^2^ has offered at room temperature an NEP of 1.1 × 10^−10^W/√Hz. For example, it is possible to improve further the photoresponse performance through a mixture of QDs with various compounds, which is also promising for the implementation of a multi-colour imager.

## Figures and Tables

**Figure 1 nanomaterials-10-00861-f001:**
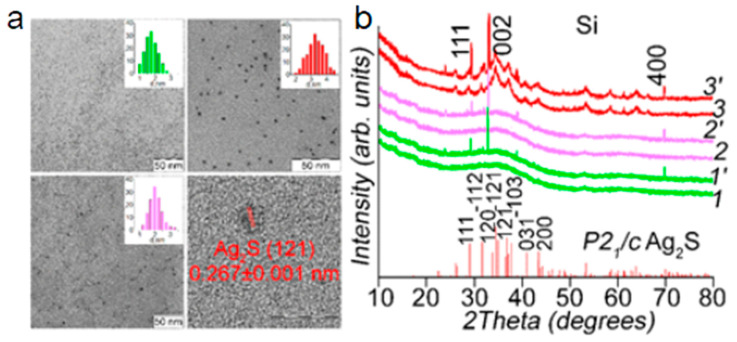
Transmission electron microscopy (TEM) images with size histograms of Ag_2_S quantum dots (QDs) deposited on an Si substrate (**a**); X-ray diffraction (XRD) patterns of Ag_2_S QDs with different sizes (1–3), curves with dashed numbers (1’–3’) respect to Ag_2_S QDs deposed on Si substrate (**b**).

**Figure 2 nanomaterials-10-00861-f002:**
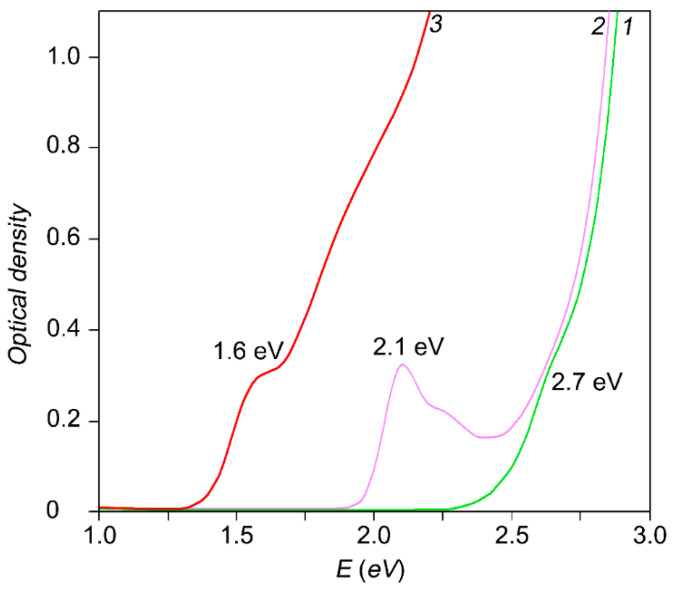
Optical absorbance spectra of Ag_2_S QDs colloidal solutions. The curves have characteristic features at of 1.6 eV, 2.1 eV, and 2.7 eV, for solutions with an Ag_2_S QDs average size of 3.2 nm, 2.0 nm, and 1.8 nm, respectively.

**Figure 3 nanomaterials-10-00861-f003:**
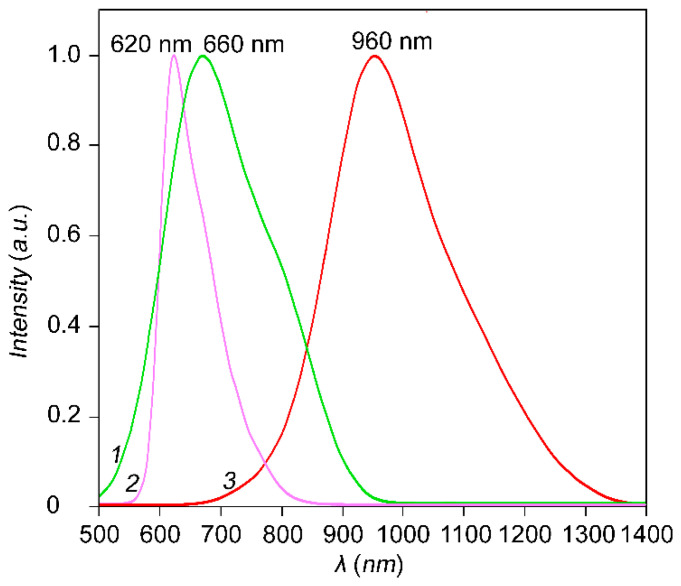
Luminescence spectra of Ag_2_S QDs colloidal solutions. The maxima of luminescence bands are at 660 nm, 620 nm, and 950 nm for Ag_2_S QDs with average size of 1.8 nm, 2.0 nm and 3.2 nm respectively.

**Figure 4 nanomaterials-10-00861-f004:**
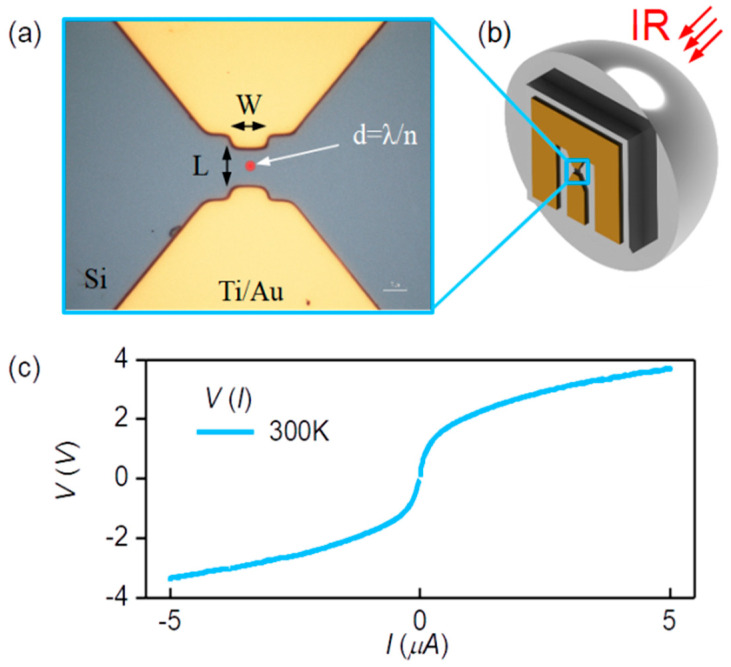
(**a**) An optical image of the inner part of the device; infrared (IR) radiation is focused on the Si surface between the Ti/Au contacts. (**b**) An image of the device mounted on the Si lens. (**c**) An IV curve of the Au/Ti/Si/Ti/Au structure; the shape of the IV curve indicates the formation of a spatial charge distribution at the Si/Ti interface.

**Figure 5 nanomaterials-10-00861-f005:**
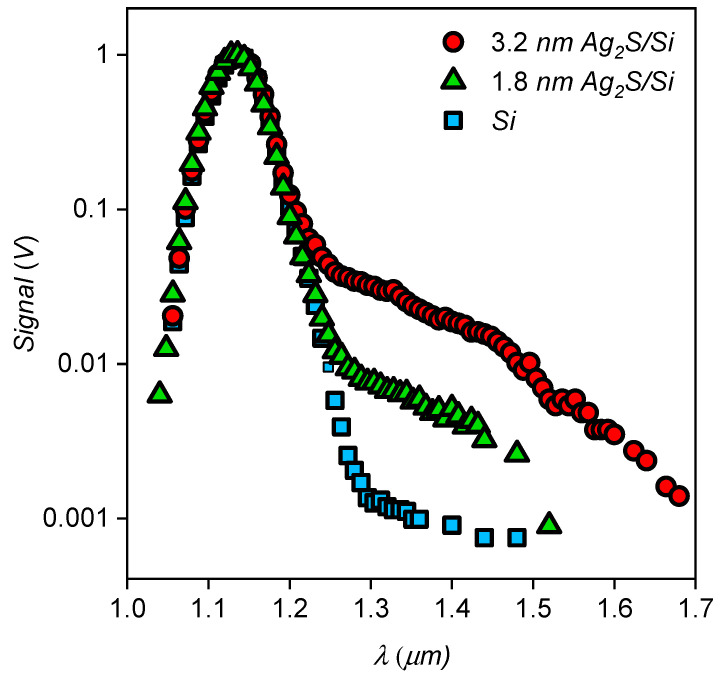
Spectral response curves of the uncovered Si and Ag_2_S/Si heterostructures. The comparison of the curves reflects a sub-band gap IR photon absorption in Si.

**Figure 6 nanomaterials-10-00861-f006:**
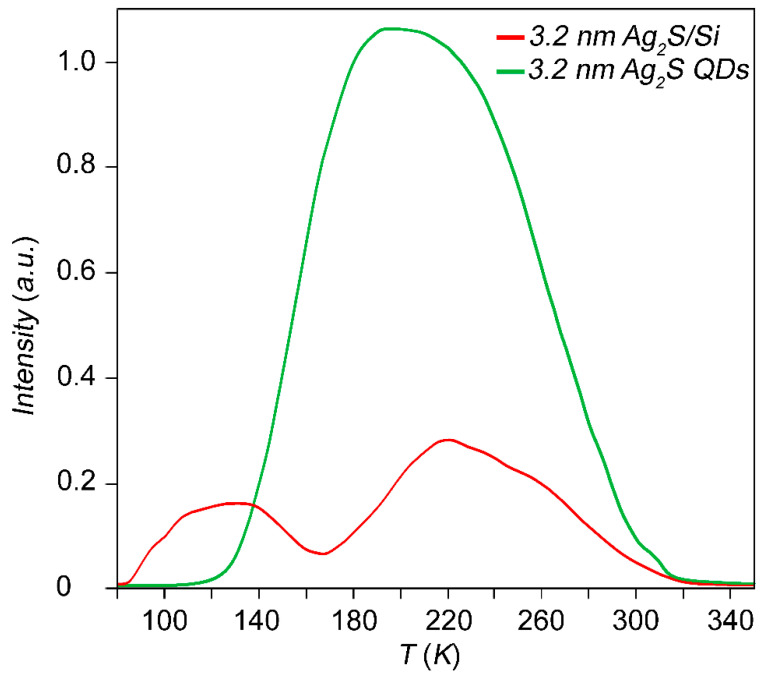
Curves of thermoluminescence for Ag_2_S QDs colloidal solutions (top curve) and Ag_2_S QDs deposed on Si substrate (bottom curve) with average size of 3.2 nm.

**Figure 7 nanomaterials-10-00861-f007:**
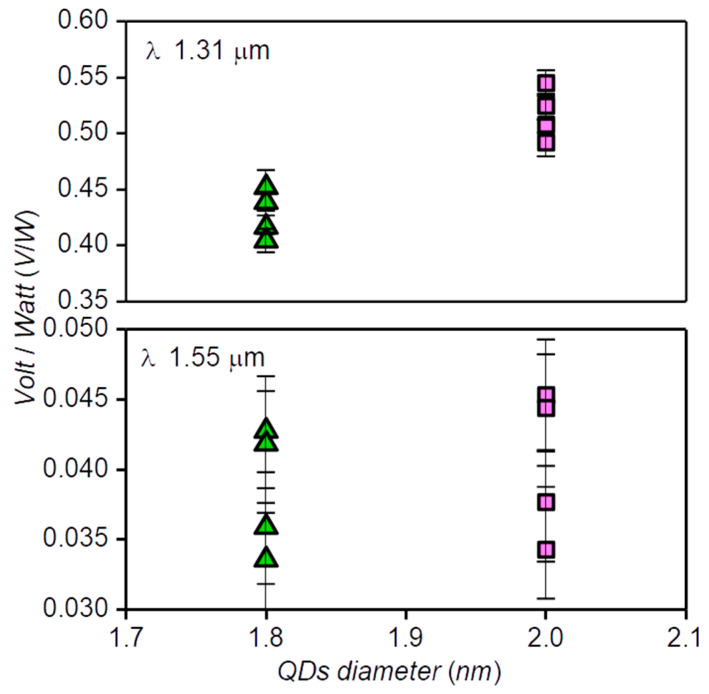
The responsivity measured for two groups of devices with QDs of diameters 1.8 and 2 nm at radiation wavelengths of 1.31 and 1.55 μm, which are shown in the upper and lower panels, respectively. The responsivity at λ 1.31 μm for heterostructures with a QD size of 2 nm is noticeably larger than that for heterostructures with a QD size of 1.8 nm. For the same devices, measurements at λ of 1.55 μm do not reveal any dependence on the size of the QD within the experimental error.

**Figure 8 nanomaterials-10-00861-f008:**
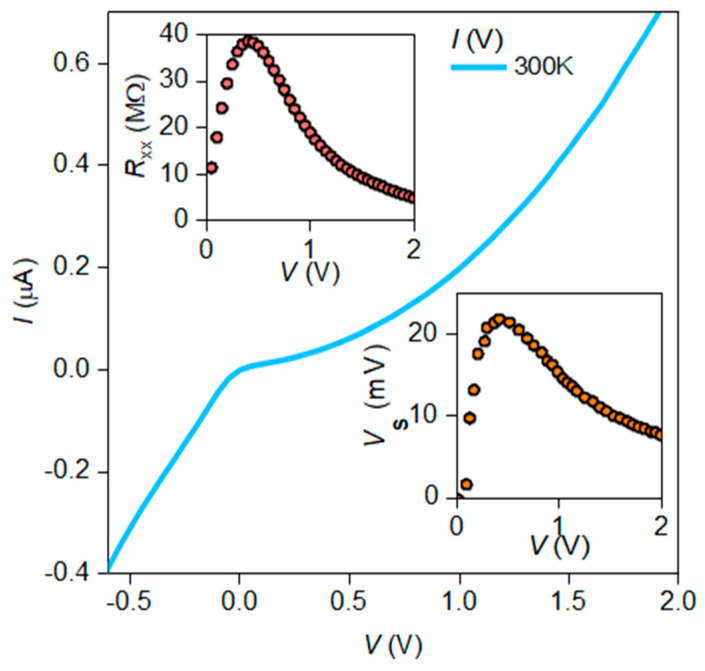
Measurements in the case of voltage-biased Ag_2_S/Si heterostructures. The IV cure under a positive potential displays a pronounced exponential character. The IV curves of the devices with different QDs were the same and depended on the quality of the Si interface under the Ti/Au contacts and their size. The deviation from a truly exponential shape is caused by the variation of the Schottky contact area, which decreases with the increase of the forward bias voltage. The top insert presents the dependence of the resistance Rxx of Ag_2_S/Si heterostructure on the applied voltage. The bottom insert reflects the dependence of the device signal on the applied voltage; the signal increases more than 20-fold in comparison with the signal at zero bias. The maximum sensitivity is achieved in the region with the maximum non-linearity of the IV curve.

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
