# Peer review of "Ag2S QDs/Si Heterostructure-Based Ultrasensitive SWIR Range Detector"

_nanomaterials, 2020, doi:10.3390/nano10050861_

Round 1

Reviewer 1 Report

The manuscript is publishable on Nanomaterials.

Author Response

We would like to thank the reviewer for the comment, we agree with the reviewer.

Reviewer 2 Report

The authors improved the manuscript significantly and made it ready for publication.  

Author Response

We would like to thank the reviewer for the comment, we agree with the reviewer.

This manuscript is a resubmission of an earlier submission. The following is a list of the peer review reports and author responses from that submission.

Round 1

Reviewer 1 Report

Shortwave infrared (SWIR) detectors have wide applications and are currently under developments. This research extended the response range of Si sensors from 1.0um to 1.5-1.7um (SWIR range) using Ag2S QDs.  The SWIR response increases significantly beyond 1.25 um. The work is interesting while needs a major reversion before publication.

1.
In the first paragraph, the authors introduced the background of their research and stated that "the problem of expanding the absorption range of Si in the far-IR is of considerable fundamental and practical importance".  The far-IR (with a wavelength of 15 - 1000 um) is not the topic (SWIR, with a wavelength of 0.7-3 um) of the manuscript.

2. In the introduction, besides chalcogen doping to create subbed absorption of Si, element As is also doped to Si for IR detections and Si:As detectors are commercially employed to detect IR with wavelength of 5-28 um. The authors should cite related publications beside S, Se, Te, Au, Ag, and Ti doping.

3. Besides GaAs and InAs non-silicon materials, HgCdTe and InSb are commercially employed to detect IR at wavelength of 0.8-2.5um and 0.8-5.4 um respectively. These information is missed in the introduction.

4. The authors deposited Ag2S QDs on Si substrates to form Ag2S/Si heterostructures. The Ag2S QDs were suspending in colloidal solutions. Usually colloidal QDs are coated with a layer of organic molecules. Detail experimental process of Ag2S synthesis and deposition should be given in the manuscript to clearly show that the heterostructure is Ag2S/Si, not Ag2S / molecule / Si in the research.

5. During the fabrication of Ag2/Si heterostructures, the deposited QD/Si was heated to 120 C. Details should be given, to clearly show that 1) the size of Ag2S does not change under the heating, 2) the Ag2S phase does not change, 3) the Si surface is not oxidized.

6. The authors claimed that the "effective absorption of SWIR photons can be either due to the QD crystal lattice defect" and "The spectral analysis ... made it possible to identify two main mechanisms behind it (photovoltaic effect): absorption of IR radiation by defects in the crystalline structure of the Ag2S QDs". So the physical properties, such as defects, crystallinity, absorbance spectrum of the Ag2S QDs should be given in the manuscript. The authors should show X-ray diffraction patterns, high-resolution TEM images etc.

7. The authors contributed the photo response to main mechanisms: "absorption of IR radiation by defects in the crystalline structure of the Ag2S QDs or due to quantum QD-induced surface states in Si". Unfortunately neither defects in Ag2S QDs nor surface states in Si was experimentally characterized in the manuscript.

8. Experimental error bar should be shown in Figure 3, to indicate the reliability of experimental values.  

9. In Figure 4, no QD size was claimed for the curve.  It is better to compare the IV curves of difference QDs. 

10. Fig 1a is very similar to Fig 1a in Ref [30]. The authors should explain improvements and differences between this manuscript and Ref [30].

11. The authors claimed that "performed a detailed study of the response of Ag2S/Si heterostructures with a QD size of 1.8 nm and 2 nm" while the data of 2nm QD was missed in Figure 2.

12. In Abstract: "experimental studies of the photovoltaic effect in Si coated with Ag2S quantum dots", it is better to say "experimental studies of the photovoltaic effect in Ag2S QD / Si heterostructures".

13. In the title, it is better to use "Ag2S QDs / Si heterostructure", not "QDs Ag2S / Si heterostructure".

14. Abbreviation should be defined in the manuscript, such as SWIR, QD, CVD etc.

15. Line 161: MM and subMM are not defined.

16. Some typos: line 70: "IV curve of one our Au ..." should be "IV curve of one Au ...". nm2 should be nm^2; Ag2S should be Ag_2S.

17. Line 186: the page number should be 4598-4810.

Reviewer 2 Report

The manuscript reports a CMOS-compatible SWIR detector with better sensitivity at 1.55 um than existing commercial devices.  The fabrication process appears straightforward and scalable.  Therefore the results would be very interesting to a large variety of applications including telecommunication and IR imaging.

However, I've ranked the merit of the manuscript rather low because I find that the claims overlap too much with a previous publication by the same authors (reference 30, line 49).

In my view, the major claims made in this paper -- that Ag2S nanoparticles can induce absorption in the Si-Ti Schottky photo-diodes in the SWIR, that the enhancement is due to electrons in the Si being excited into defect stages of the Ag2S, and that  a high sensitivity of 10^11 cm Hz-2 W-1 at 1.55 um have all been reported by the authors in reference 30. In fact, I had written down a list of points that I felt the authors should address in the manuscript and found the answers to most of them in reference 30.  Consequently, I find it inappropriate that this previously published article is mentioned obscurely at the end of the intro.  Instead I believe the article should reference it often and reduce the amount of words dedicated to the fabrication and operation of the devices, as both subjects were better explained in reference 30.  (or reference the previous article and put the redundant information in supplementary, if required by the journal)

Compared to reference 30, the new finding in the present work is that enhancement of the detectors' response in at 1.3 um is related to the size of the deposited nanoparticles.  Two different particles sizes were used in this work compared to reference 30.  

The fact that there is enhancement at 1.5um and that it is independent of nanoparticle size confirms finding in reference 30.

In conclusion, I believe the authors have made an interesting device with many application potentials.  However I believe the new finding in this manuscript to be too incremental from its predecessor for publication as it is. Also, a revised manuscript should be rewritten to emphasize the progress made in this work as compared to previous work.

Reviewer 3 Report

The authors claim a photovoltaic effect in near infrared basen on a Si-Ag2S(QD) heterostructure. The subject is of interest but there are too few results presented to support the scientific claims. The major point stands on QD's with distribution of diameters at 1.8, 2 and 3.2 nm. However, no data are presented to support this critical claim. Overall, the presented data are difficult to support even a journal letter paper (only three figures with results).